# *Ixeris polycephala* Extract Alleviates Progression of Benign Prostatic Hyperplasia via Modification of Proliferation, Apoptosis, and Inflammation

**DOI:** 10.3390/ph17081032

**Published:** 2024-08-05

**Authors:** Eun-Bok Baek, Youn-Hwan Hwang, Eun-Ju Hong, Young-Suk Won, Hyo-Jung Kwun

**Affiliations:** 1Department of Veterinary Pathology, College of Veterinary Medicine, Chungnam National University, Daejeon 34134, Republic of Korea; baekeunbok@hanmail.net (E.-B.B.); bioeun59@naver.com (E.-J.H.); 2Herbal Medicine Research Division, Korean Institute of Oriental Medicine, Daejeon 34054, Republic of Korea; hyhhwang@kiom.re.kr; 3Laboratory Animal Resource Center, Korean Research Institute of Bioscience and Biotechnology, Chungbuk 34141, Republic of Korea; yswon@kribb.re.kr

**Keywords:** *Ixeris polycephala*, proliferation, apoptosis, inflammatory response

## Abstract

Benign prostatic hyperplasia (BPH) is a urogenital disorder that is common in aging men. *Ixeris polycephala* (IP) is used in traditional medicine and contains pharmacologically active compounds. However, the effect for BPH progression has not been elucidated. We herein examined the protective potential of IP extract on a testosterone-induced model of BPH in rats. To generate the BPH model, daily subcutaneous administration of testosterone was applied for 4 weeks. During this period, the rats were also administered a daily oral gavage of IP (150 mg/kg), finasteride (positive control), or vehicle. Testosterone treatment was associated with a significantly higher prostate-to-body weight ratio, serum dihydrotestosterone (DHT) level, and prostatic gene expression of 5α-reductase compared to untreated controls. Notably, IP plus testosterone co-treatment was associated with decreased epithelial thickness, down-regulation of proliferating cell nuclear antigen (PCNA) and cyclin D1, and upregulation of pro-apoptotic signaling molecules. IP co-treatment also down-regulated pro-inflammatory cytokines, cyclooxygenase-2 (COX-2), and inducible nitric oxide synthase (iNOS) and decreased inflammatory cell infiltration compared to the levels seen in the testosterone-induced BPH. IP appears to protect rats against the progression of testosterone-induced BPH by alleviating prostate cell growth and inflammatory responses, and thus may have potential for clinical use against BPH progression.

## 1. Introduction

Benign prostatic hyperplasia (BPH) is the most frequent disease among men older than 60 years; it is found in 30–40% of men overall and in almost 70% those older than 80 years [1]. BPH is characterized by prostate enlargement and lower urinary tract symptoms (LUTS) that typically require medical consultation and decrease quality of life (QOL) [1].

The interstitial and molecular mechanisms underlying BPH are not fully understood, but they are believed to involve oxidative stress, environmental stress, inflammatory mediators, hormones, and dietary factors [1,2]. The pathogenesis of BPH reportedly involves the actions of androgens (e.g., testosterone and dihydrotestosterone (DHT)) in stimulating prostatic growth, as well as contributions by 5α-reductase [1]. A lack of balance between the cell cycle and apoptotic cell death has been suggested to increase cell numbers and lead to out-of-control prostatic cell growth [3]. Chronic inflammation was found alongside histological changes in prostatic tissues from BPH patients [4]. In the development of BPH, bacterial infection may trigger local inflammation and the related secretion of growth factors, cytokines, and chemokines, leading to proliferation of epithelial and stromal prostatic cells [4]. This inflammatory response is thought to be perpetuated by prostatic self-antigens that are released upon tissue damage to both stimulate the autoimmune system and sensitize the immune response. Prostatic stromal cells are key to this process, as they may function to activate CD4^+^ lymphocytes and upregulate pro-inflammatory cytokines and chemokines [5].

At present, BPH treatment mainly aims to ameliorate symptoms (e.g., LUTS), inhibit disease progression, reduce complications, and improve QOL [6]. The various drugs currently used to treat BPH include alpha1-adrenergic antagonists (alfuzosin, doxazosin, terazosin, and tamsulosin), β3-adrenoceptor agonists (vibegron and mirabegron), 5α-reductase inhibitors (finasteride and dutasteride), muscarinic receptor antagonists (tolterodine), phosphodiesterase 5 inhibitors (tadalafil), and traditional medicines such as herbal extracts.

The relevant herbs include *Ixeris polycephala* (IP); this is a member of the genus *Ixeris Cass.* and is mainly distributed in Europe and Asia [7,8]. The genus *Ixeris* was believed to be rich in bioactive substances including sesquiterpene lactones, triterpenes and steroids, phenylpropanoids, phenols, amino acids, and fatty alcohol/acids [9]. IP is reported to have various bioactive compounds including steroids, triterpenes, sesquiterpene, lactones, amino acids, and fatty acids [10]. Previous pharmacological studies have revealed that some identified chemicals from *Ixeris* exhibited diverse bioactivities, such as anti-inflammatory, neuroprotective, hepatoprotective, and antimicrobial activities [9]. However, the full range of IP-derived phytochemicals and their potential activities for prostatic disease remain to be explored in detail. In the present study, we explored the efficacy of IP extract in the management and treatment of BPH. The components of IP extract were identified and the mechanisms of the protective effect were examined.

## 2. Results

### 2.1. Ultra-Performance Liquid Chromatography–Tandem Mass Spectrometry (Uplc-Ms/MS) Analyses of IP

UPLC-MS/MS analysis was conducted to identify IP constituents. For the 10 compounds identified (Figure 1), the mass spectral characteristics are presented in Table 1 and compared with the retention time, precursor ion, and MS/MS fragments of each authentic standard.

### 2.2. Prostatic Weight, Dht Level, and 5α-Reductase Level in a Rat Model of Bph

To examine the therapeutic potential of IP against BPH, we generated a rat model of testosterone-induced BPH and assessed prostate weight. There was no animal death during the experimental period, and the body weight and absolute prostate weight of each group are shown in Table 2. Compared to normal control (NC) animals, testosterone-treated animals exhibited significant increases in prostate-to-body weight. These increases were significantly attenuated in the finasteride (FIN) and IP groups (Table 2, Figure 2A). Testosterone is converted to DHT via 5α-reductase, and an increased level of DHT is a critical pathogenic mediator of BPH [11,12]. Here, we found that the serum DHT level was higher in the BPH group compared to the NC group, whereas this enhancement was significantly rescued in the FIN and IP groups (Figure 2B). Consistently, the mRNA expression levels of 5 alpha-reductase 2 (*Srd5a2*) were significantly reduced in the FIN and IP groups compared to the BPH group (Figure 2C). These results indicate that IP decreases the prostatic weight of BPH model rodents at least partially through decreasing *Srd5a2* expression and subsequent DHT production.

### 2.3. Histological Analysis of Prostatic Tissues from Bph Rats

Next, we performed histological analysis to study the morphological change of prostatic tissues. The morphology of these tissues was normal in the NC group, whereas samples from the BPH group showed evidence of glandular hyperplasia, enhanced epithelial thickness, and decreased glandular luminal area (Figure 3A). In the FIN and IP groups, in contrast, the epithelial thickness was markedly lower than that in the BPH group (Figure 3B). This indicates that IP effectively decreases epithelial hyperplasia in prostatic tissues of BPH rats.

### 2.4. Prostatic Cell Proliferation in BPH Rats

Since an imbalance between proliferation and apoptosis is a key aspect of BPH pathogenesis [13], we used proliferating cell nuclear antigen (PCNA) staining to examine how IP impacts proliferation. PCNA^+^ cells were significantly increased in the BPH group compared to the NC group, but this increase was reversed in the FIN and IP groups (Figure 4A,B). The protein expression of cyclin D1 was also notably enhanced in the BPH group, whereas this change was significantly suppressed in the FIN and IP groups (Figure 4C). These data suggest that IP prevents the progression of BPH at least partly by exerting anti-proliferative activity.

### 2.5. Prostatic Cell Apoptosis in BPH Rats

To evaluate the effect of IP on apoptosis in our experimental setting, we used Western blotting to analyze apoptotic proteins. The pro-apoptotic markers, cleaved caspase-3, and bcl-2-associated X protein (Bax) were significantly increased in the FIN and IP groups vs. the BPH group (Figure 5A,B). The anti-apoptotic protein and B-cell lymphoma 2 (Bcl-2) was increased in prostates of BPH rats but recovered in the FIN and IP groups (Figure 5C). These data indicate that IP promotes apoptosis by modulating caspase-3, Bax, and Bcl-2-dependent signaling.

### 2.6. Inflammatory Cytokines in BPH Rats

Inflammation is thought to contribute to BPH pathogenesis [5]. Here, we found that prostatic tissue morphology was normal (i.e., having few inflammatory cells) in the NC group, showed increased inflammatory cells in the prostatic interstitial lesions of BPH group animals, and exhibited rescue (resembling NC samples) in the FIN and IP groups (Figure 6A). Our RT-qPCR analysis of prostatic tissues supported these histological observations: the relative expression levels of the cytokines, interlukin-6 (*Il6*) and *Il8*, were increased in prostatic tissues of the BPH group, and these increases were significantly alleviated in the FIN and IP groups (Figure 6B,C). These results show that IP attenuates testosterone-induced inflammation and cytokine production in prostatic tissues of BPH rats.

### 2.7. Cyclooxygenase-2 (Cox-2) and Inducible Nitric Oxide Synthase (Inos) Expression in Bph Rats

COX-2 and iNOS are critically connected to inflammation with prostate growth [5]. Here, we found that the protein expression levels of COX-2 and iNOS were higher in the BPH group vs. the NC group, but these increases were significantly attenuated in the FIN and IP groups (Figure 7A,B). Consistently, the relative mRNA expression levels of *Cox2* and *Nos2* were markedly inhibited in the FIN and IP groups (Figure 7C,D).

## 3. Discussion

We herein examined the potential therapeutic benefit of IP against BPH in a testosterone-induced rat model. Our results revealed that the testosterone-induced enhancements of the prostate-to-body weight ratio, serum DHT level, and prostatic *Srd5a2* expression level were significantly decreased in IP-treated rats. Administration of IP to BPH rats also alleviated prostatic epithelial cell proliferation, altered the expression patterns of pro- and anti-apoptotic proteins, and ameliorated the increases in inflammatory cell infiltration and inflammatory cytokines. These results collectively suggest that IP prevents testosterone-induced BPH progression in this rat model.

In the prostate, 5α-reductase acts upon testosterone to produce DHT, which is an androgen that is responsible for prostate growth [14] and has important functions in BPH pathogenesis and progression [12]. DHT binds to the androgen receptor with high affinity, and this complex initiates transcription and ultimately controls the regulation of the prostatic cell cycle, cell growth, and differentiation, contributing to prostatic disease progression [15]. Prostate DHT levels can be reduced with drugs that inhibit 5α-reductase, and this strategy is currently used to treat BPH in clinical settings [16]. In the present work, we show that administration of FIN (positive control) or IP treatment decreases the serum level of DHT and the prostatic mRNA expression of *Srd5a2* in prostatic tissues, showing an inhibitory effect on 5α-reductase expression.

Since a lack of balance between prostatic cell proliferation and apoptosis is considered to be important to BPH pathogenesis [13], we analyzed these parameters in our experimental setting by examining the levels of PCNA positivity and protein expression levels of proliferation- and apoptosis-related molecules. PCNA is involved in DNA replication, DNA damage repair, and cell cycle progression and critically contributes to pathological conditions such as BPH [17,18]. Cyclin D1 critically regulates proliferation by linking extracellular signals with cell cycle progression [19]; its expression is very sensitive to proliferative signaling [20]. Bax is a pro-apoptotic protein that triggers mitochondrial cytochrome c release and thereby activates the critical apoptotic enzyme caspase-3 [21]. Consistent with the results described above, administration of IP to testosterone-treated rats notably decreased the number of PCNA^+^ cells and the expression levels of cyclin D1 and Bcl-2, while increasing those of cleaved caspase-3 and Bax in prostate tissues. Thus, IP appears to inhibit proliferation and enhance apoptosis in the prostatic tissues of BPH model rats, potentially by exerting beneficial effects on the cell cycle.

Prostatic inflammation is suggested to be critically involved in prostate enlargement and BPH progression [22]. Debate remains regarding the origin of this inflammation, although it is generally expected to be multifactorial [23]. In the prostate, both acute and chronic inflammation result in the accumulation of immunocompetent cells, mainly T lymphocytes and macrophages. Under acute inflammation, epithelial and stromal cells release IL-6; consistent with the belief that BPH critically involves inflammation, the IL-6 level is higher in BPH tissues compared to normal prostate tissues [24]. The pro-inflammatory cytokine, IL-8, is secreted by prostate epithelial cells and contributes to leukocyte chemotaxis. BPH tissue samples are characterized by increased IL-8, which is highest among men with BPH plus chronic prostatitis [25]. Consistent with the prior findings, we observed that inflammatory cell infiltration in prostatic stroma and the mRNA expression levels of *Il6* and *Il8* were significantly increased in the BPH group, but these changes were significantly reversed with IP treatment.

iNOS is an enzyme that directs the formation of reactive nitrogen oxide, which can cause damage to prostate cells [26] and contribute to prostate tumorigenesis [26]. COX-2 expression was upregulated in prostate luminal epithelial cells of BPH patients, and a COX-2 inhibitor reduced cell proliferation and increased apoptosis kevel in BPH rats [27,28]. Here, we report that the protein expression levels of iNOS and COX-2 were increased in the BPH group relative to the NC group, but this change was ameliorated via IP treatment. These results suggest that IP extract exhibits a protective effect via anti-inflammatory activities.

We herein separated and identified 10 compounds from IP extract, and the major constituents were considered to be cynaroside, apigenin, acacetin, luteolin, chlorogenic acid, and arctigenin. Cynaroside exerted antibacterial, antifungal, anti-inflammatory, and anticancer effects in various experimental systems [29]. Apigenin is a flavonoid that was shown to attenuate prostatic cell proliferation in human prostatic stromal cells, possibly through cell cycle arrest [30]. Acacetin exhibited diverse pharmacological activities, such as neuroprotective, cardioprotective, anticancer, and antimicrobial properties [31]. Luteolin and arctigenin show anti-inflammatory activity [32,33]. Chlorogenic acid reportedly exhibited 5α-reductase inhibitory activity and a protective effect against BPH in vivo [34]. In addition, vanillic acid was reported to protect rats against testosterone-induced BPH and inhibit prostatic epithelial cell proliferation [35]. The multiple constituents from IP extracts might exhibit biological and pharmacological activity, and some components, including polyphenol, may contribute to its ability to protect against BPH. In our analysis results, some peaks acquired through MS fragmentation were not identified in Figure 1. It is necessary to confirm the findings more clearly through the separation and purification study of IP phytochemicals. In a previous study, IP was extracted in aqua methanol and aqua acetone [10]. Depending on extraction solvent type or extraction method, extraction yield, the amount of phytochemical constituents, and their antioxidant activity could be different [10].

## 4. Materials and Methods

### 4.1. IP Extract

IP extract was purchased from The Korea Plant Extract Bank at the Korea Research Institute of Bioscience and Biotechnology (Daejeon, Republic of Korea).

### 4.2. UPLC-MS/MS Analysis of IP

UPLC-MS/MS analyses were performed as previously described [36]. Briefly, IP and reference standards were dissolved in methanol and filtered using a syringe filter (0.2 μm pore size). IP were analyzed using a Dionex UltiMate 3000 system equipped with a Thermo Q-Exactive mass spectrometer (Thermo Fisher Scientific, Bremen, Germany). The chromatographic separation was performed on an Acquity BEH C18 column (100 × 2.1 mm, 1.7 µm) at 40 °C. The mobile phase consisted of acetonitrile (phase A) and 0.1% formic acid in water (phase B), using a gradient elution of 97% B at 0–1 min; 97–85% B at 1–2 min; 85–50% B at 2–13 min; 50–0% B at 13–20 min; 0% B at 20–23 min; 0–97% B at 23–23.5 min; and 97% B at 23.5–27.5 min. The flow rate was maintained at 0.25 mL/min, and the injection volume was 3 μL. The Q-Exactive mass spectrometer was equipped with a heated electrospray ionization source (HESI) and operated in positive/negative ion-switching modes. The ionization source was optimized using the following parameters: capillary gas temperature, 320 °C; spray voltage, 3.0 kV; S-lens RF level, 60; sheath gas flow rate, 40 arbitrary units (au); and auxiliary gas flow rate, 5 au. The mass spectra were acquired using full MS and ddMS2 scan modes. The full MS acquisition parameters were set as follows: resolution, 70,000; AGC target, 1e6; maximum IT, 100 ms; and scan range, 100–1500 *m*/*z*. The MS2 acquisition parameters were as follows: resolution, 17,500; AGC target, 1e5; maximum IT, 50 ms; loop count, 10; MSN count, 1; and normalized collision energy (NCE), 25. Data acquisition and analysis were performed using Xcalibur 4.2 and TraceFinder 4.1 (Thermo Fisher Scientific).

### 4.3. Animal Experiment

Male Sprague-Dawley (SD) rats (7-weeks-old; 150–220 g; Orient Bio, Seongnam-si, Republic of Korea) were used. During the experimental period, rats were kept under 50 ± 5% relative humidity and 22 ± 2 °C with a 12 h light/dark cycle and provided ad libitum access to sterilized tap water and normal rodent chow. Rats were randomly separated into four groups and treated as follows: (1) NC group animals were administered daily treatments of phosphate buffered saline (PBS) via oral gavage and corn oil via subcutaneous (s.c.) injection. (2) BPH group animals were administered daily treatments of PBS via oral gavage and testosterone propionate (3 mg/kg; Tokyo Chemical Industry, Tokyo, Japan) via s.c. injection. (3) FIN group animals (positive control) were administered daily treatments of finasteride (10 mg/kg, Sigma-Aldrich, Burlington, MA, USA) via oral gavage and testosterone propionate via s.c. injection. (4) IP group animals were administered daily treatments of IP (150 mg/kg) via oral gavage and testosterone propionate via s.c. injection. All treatments lasted for 4 weeks; the administration volumes were 5 mL/kg for oral doses (PBS, finasteride, and IP) and 3 mL/kg for s.c. injection. After a 4-week treatment, prostatic tissues were isolated and weighed.

### 4.4. Enzyme Linked Immunosorbent Assay (ELISA) of DHT

The serum levels of DHT were measured using a commercially available ELISA kit (ALPCO Diagnostics, Salem, NH, USA) as previous described [36].

### 4.5. Histological Examination

Prostatic tissues fixed in 10% buffered formalin solution were subjected to hematoxylin and eosin (H&E) staining as previously described [37]. Epithelial thickness was measured in prostate ventral lobes as previously described [38]. Briefly, 30 regions randomly selected from each prostate were measured in micrometers (µm) using the NIS-element software (BR5.11, Nikon, Tokyo, Japan).

### 4.6. Immunohistochemical (IHC) Staining

PCNA staining was performed using IHC techniques. Five ventral prostate lobe sections were randomly selected from each rat and incubated overnight with anti-PCNA (Abcam, Cambridge, UK) and then incubated for 1 h with biotinylated secondary antibodies. Binding was visualized by staining with diaminobenzidine (DAB; Vector Laboratories, Newark, CA, USA) and counter-staining with hematoxylin. The numbers of PCNA^+^ cells are expressed as a percentage of total cells.

### 4.7. Western Blotting

SDS-PAGE was used to resolve equal amounts of total proteins, and the bands were transferred to polyvinylidene difluoride (PVDF) membranes. The membranes were incubated overnight with anti-cyclin D1, anti-caspase-3, anti-COX-2, anti-iNOS (Cell Signaling Technology, Danvers, MA, USA), anti-Bax, anti-Bcl-2 (Santa Cruz, Dallas, TX, USA), and anti-β-actin (Sigma-Aldrich, Burlington, MA, USA). The membranes were then incubated for 2 h at room temperature with an appropriate secondary antibody, and the results were developed using a chemiluminescence detection kit and quantified using CSAnalyzer4 (Atto, Tokyo, Japan). Full-length blots/gels are presented in the Appendix A.

### 4.8. Quantitative Real-Time PCR (RT-qPCR)

Total RNA was extracted from prostate samples using the TRIzol reagent and equal amounts (1 µg) were subjected to reverse transcription (RT) using an RT kit (Toyobo, Osaka, Japan). Quantitative real-time PCR was performed with SYBR Green Master Mix (Thermo Fisher Scientific, Waltham, MA, USA) and the following primers: *Srd5a2*, ATTTGTGTGGCAGAGAGAGG (forward [f]) and TTGATTGACTGCCTGGATGG (reverse [r]); *Cox2*, CGACCTTGCTCACTTTGTTG (f) and CTCTTGCTCTGGTCAATGG (r); *Il6*, TAGTCCTTCCTACCCCAACT (f) and TTGGTCCTTAGCCACTCCTT (r); *Il8*, CATTAATATTTAACGATGTGGATGCGTTTCA (f) and GCCTACCATCTTTAAACTGCACAAT (r); *Nos2*, CTGAAGCACTAGCCAGGGAC (f) and CAAATGTGCTTGTCACCACC (r); and *Gapdh*, ACAGCAACAGGGTGGTGGAC (f) and TTTGAGGGTGCAGCGAACTT (r). Data were generated and analyzed with a 7500 Real-Time PCR System and its software (v2.3, Applied Biosystems, Waltham, MA, USA). The 2^−ΔΔCt^ method [39] was used to calculate the fold-change relative to the endogenous constitutive control (*Gapdh*).

### 4.9. Statistical Analysis

Values are expressed as mean ± standard deviation (SD). For multiple comparisons, one-way ANOVA followed by Dunnett’s post hoc test was used; *p* < 0.05 was considered significant. Statistical analyses were performed using GraphPad Prism 6 (GraphPad 6 Software, San Diego, CA, USA).

## 5. Conclusions

The present results demonstrate that IP treatment effectively decreases prostate enlargement and related histological changes in a testosterone-induced rat model of BPH. Mechanistically, IP decreases the levels of serum DHT, prostatic *Srd5a2* transcripts, cell proliferation, and inflammatory cytokines. Collectively, these findings suggest that IP could potentially be developed as a therapeutic to prevent BPH.

## Figures and Tables

**Figure 1 pharmaceuticals-17-01032-f001:**
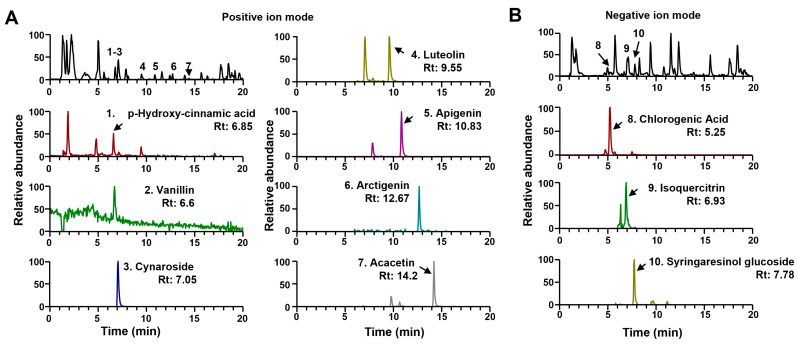
UPLC-QTOF/MS chromatograms of IP Base peak- and extracted-ion chromatograms of IP in positive (**A**) and negative (**B**) ion modes using UPLC-MS/MS. Rt, retention time.

**Figure 2 pharmaceuticals-17-01032-f002:**
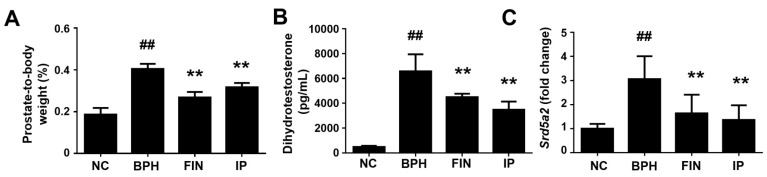
Effects of IP on prostate-to-body weight, dihydrotestosterone level, and *Srd5a2* expression in a rat model of testosterone-induced BPH. (**A**) Prostate-to-body weight. (**B**) Serum levels of DHT, examined using ELISA. (**C**) Relative *Srd5a2* mRNA levels in prostate tissue, as analyzed with RT-qPCR. Groups: NC, rats were injected with corn oil and treated with PBS; BPH, rats were injected with TP (3 mg/kg) and treated with PBS; FIN, rats were injected with TP (3 mg/kg) and treated with finasteride (10 mg/kg); IP, rats were injected with TP (3 mg/kg) and treated with IP (150 mg/kg). Data are expressed as mean ± SD. ^##^ *p* < 0.01 vs. the NC group; ** *p* < 0.01 vs. the BPH group.

**Figure 3 pharmaceuticals-17-01032-f003:**
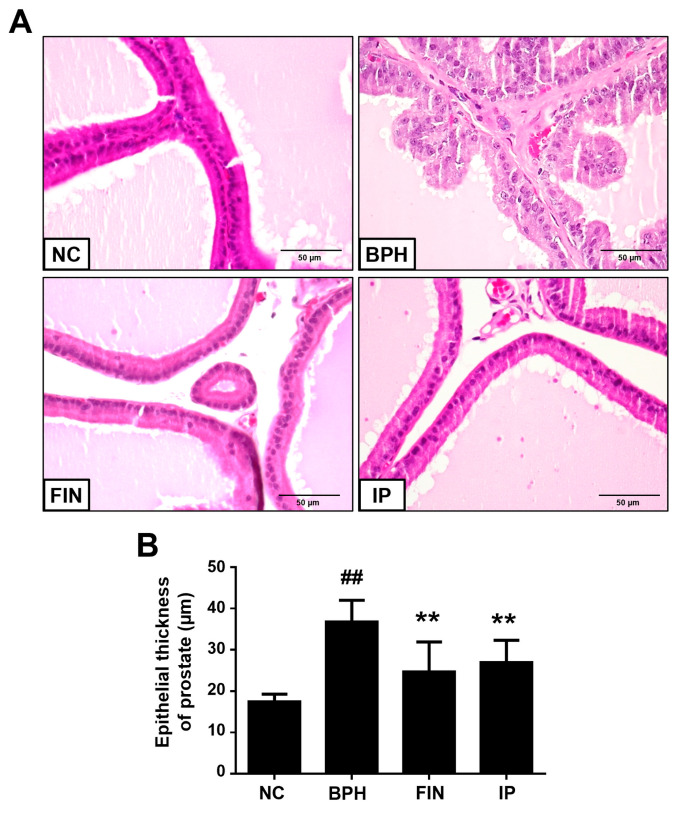
Effects of IP on prostatic histological changes in BPH rats. (**A**) Representative H&E-stained prostate tissues (magnification, ×400). (**B**) The epithelial thickness of prostate tissues. Groups: NC, rats were injected with corn oil and treated with PBS; BPH, rats were injected with TP (3 mg/kg) and treated with PBS; FIN, rats were injected with TP (3 mg/kg) and treated with finasteride (10 mg/kg); IP, rats were injected with TP (3 mg/kg) and treated with IP (150 mg/kg). Data are expressed as mean ± SD. ^##^ *p* < 0.01 vs. the NC group; ** *p* < 0.01 vs. the BPH group.

**Figure 4 pharmaceuticals-17-01032-f004:**
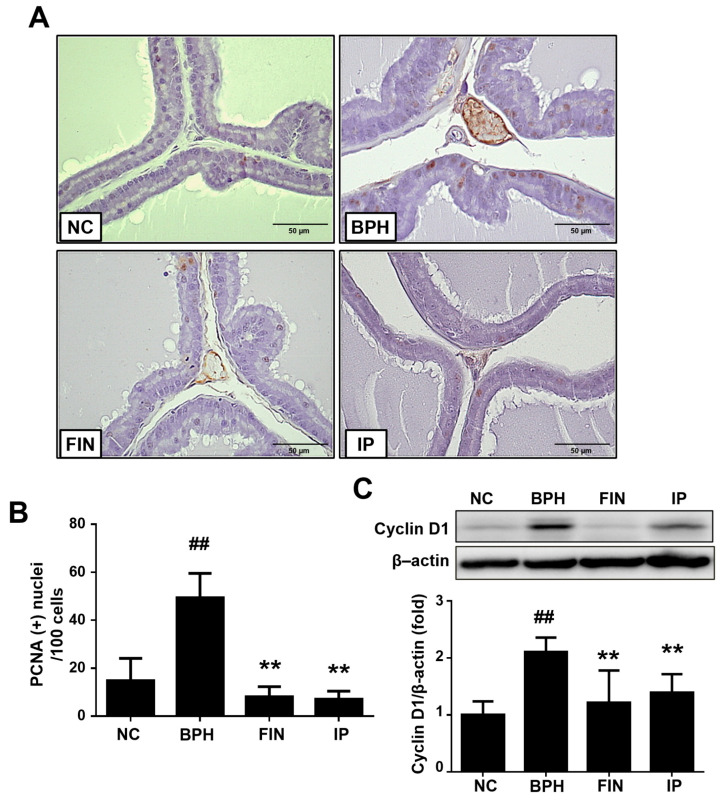
Effects of IP on prostatic cell proliferation. (**A**) Representative images showing immunohistochemical analysis of PCNA (magnification, ×400). (**B**) Numbers of PCNA^+^ cells in the prostate. (**C**) Western blot analysis of cyclin D1 expression, with β-actin used as a normalized control. Groups: NC, rats were injected with corn oil and treated with PBS; BPH, rats were injected with TP (3 mg/kg) and treated with PBS; FIN, rats were injected with TP (3 mg/kg) and treated with finasteride (10 mg/kg); IP, rats were injected with TP (3 mg/kg) and treated with IP (150 mg/kg). Data are expressed as the mean ± SD. ^##^ *p* < 0.01 vs. the NC group; ** *p* < 0.01 vs. the BPH group.

**Figure 5 pharmaceuticals-17-01032-f005:**
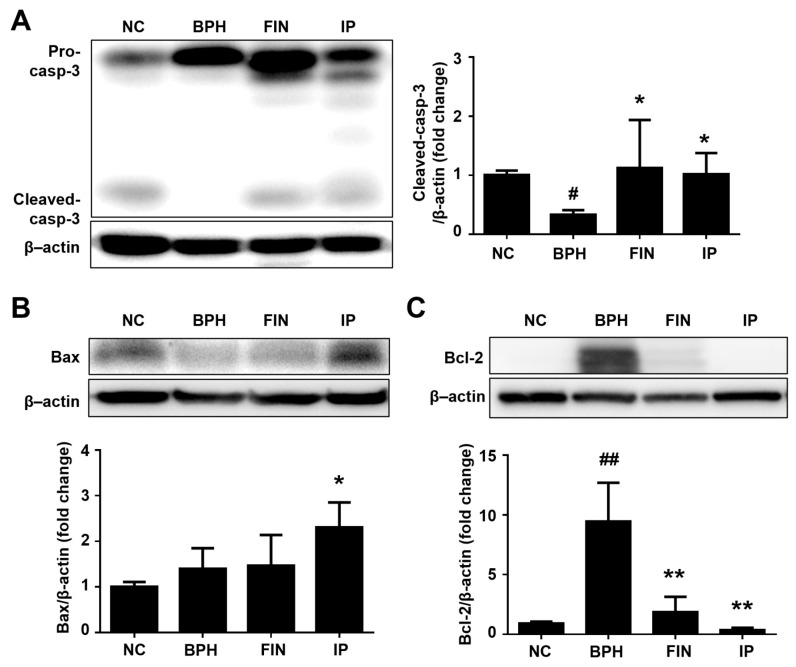
Effects of IP on apoptotic protein expression in prostatic tissues. The protein expression levels of pro- and cleaved caspase-3 (**A**), Bax (**B**), and Bcl-2 (**C**) were determined via Western blotting, with β-actin used as a normalization control. Groups: NC, rats were injected with corn oil and treated with PBS; BPH, rats were injected with TP (3 mg/kg) and treated with PBS; FIN, rats were injected with TP (3 mg/kg) and treated with finasteride (10 mg/kg); IP, rats were injected with TP (3 mg/kg) and treated with IP (150 mg/kg). Data are expressed as mean ± SD. ^#^ *p* < 0.05 and ^##^
*p* < 0.01 vs. the NC group; * *p* < 0.05 and ** *p* < 0.01 vs. the BPH group.

**Figure 6 pharmaceuticals-17-01032-f006:**
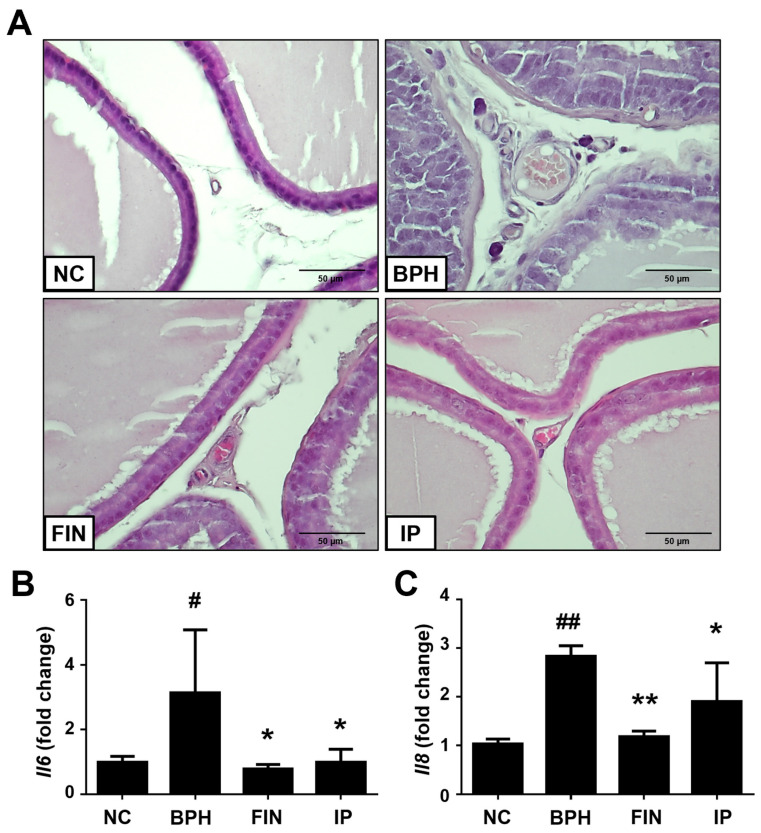
Effects of IP on prostatic inflammatory cytokines. (**A**) Representative images of prostate tissues, with rats of the BPH group showing stromal infiltration of inflammatory cells (magnification, ×400). (**B**,**C**) mRNA expression levels of Il6 (**B**) and Il8 (**C**) in prostate tissue, as assessed with RT-qPCR. Groups: NC, rats were injected with corn oil and treated with PBS; BPH, rats were injected with TP (3 mg/kg) and treated with PBS; FIN, rats were injected with TP (3 mg/kg) and treated with finasteride (10 mg/kg); IP, rats were injected with TP (3 mg/kg) and treated with IP (150 mg/kg). Data are expressed as mean ± SD. ^#^ *p* < 0.05 and ^##^
*p* < 0.01 vs. the NC group; * *p* < 0.05 and ** *p* < 0.01 vs. the BPH group.

**Figure 7 pharmaceuticals-17-01032-f007:**
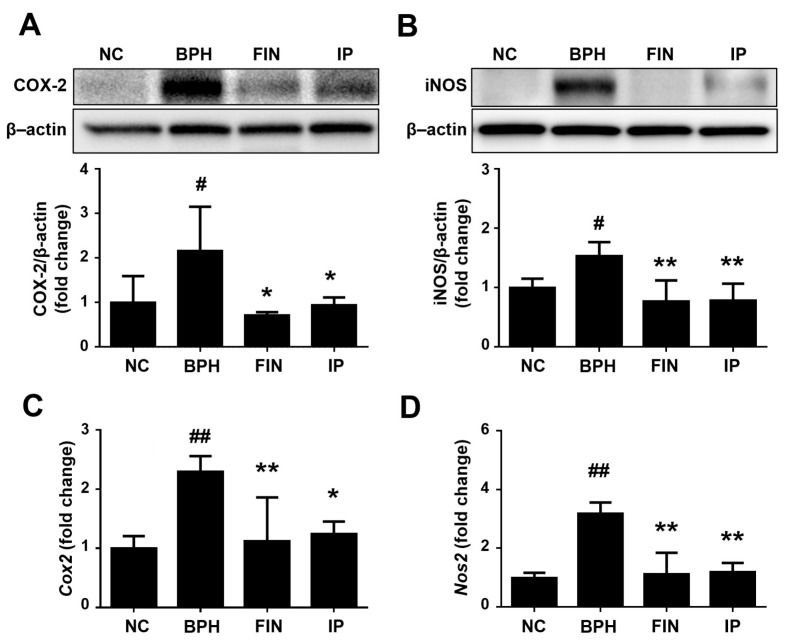
Effects of IP on the expressions of COX-2 and iNOS in prostatic tissue. (**A**,**B**) The protein expression levels of COX-2 (**A**) and iNOS (**B**) were determined using Western blotting; β-actin was used as a normalized control. (**C**,**D**) RT-qPCR was performed to analyze the mRNA levels of *Cox2* (**C**) and *Nos2* (**D**). Groups: NC, rats were injected with corn oil and treated with PBS; BPH, rats were injected with TP (3 mg/kg) and treated with PBS; FIN, rats were injected with TP (3 mg/kg) and treated with finasteride (10 mg/kg); IP, rats were injected with TP (3 mg/kg) and treated with IP (150 mg/kg). Data are expressed as mean ± SD. ^#^ *p* < 0.05 and ^##^ *p* < 0.01 vs. the NC group; * *p* < 0.05 and ** *p* < 0.01 vs. the BPH group.

**Table 1 pharmaceuticals-17-01032-t001:** Phytochemicals identified in the methanolic extract of IP via ultra-performance liquid chromatography–tandem mass spectrometry (UPLC-MS/MS).

No.	Identification	Formula	R_t_ (min)	Adduct	Calculated (*m*/*z*)	Measured (*m*/*z*)	Error (ppm)	MS/MS Fragments (*m*/*z*)	Measured Area (×10^8^)
1	p-Hydroxy-cinnamic acid	C_9_H_8_O_3_	6.58	[M+H]^+^	165.0546	165.0548	1.333	147, 84	0.10
2	Vanillin	C_8_H_8_O_3_	6.67	[M+H]^+^	153.0546	153.0547	0.740	153, 125, 133, 93, 72	0.08
3	Cynaroside	C_21_H_20_O_11_	7.05	[M+H]^+^	449.1078	449.1078	−0.048	287	28.65
4	Luteolin	C_15_H_10_O_6_	9.55	[M+H]^+^	287.0550	287.0549	−0.379	133	2.06
5	Apigenin	C_15_H_10_O_5_	10.83	[M+H]^+^	271.0601	271.0600	−0.378	159	7.40
6	Arctigenin	C_21_H_24_O_6_	12.67	[M+H]^+^	373.1646	373.1646	0.057	305, 237, 177, 137	1.48
7	Acacetin	C_16_H_12_O_5_	14.20	[M+H]^+^	285.0758	285.0756	−0.554	130	4.07
8	Chlorogenic Acid	C_16_H_18_O_9_	5.25	[M−H]^−^	353.0878	353.0870	−2.278	191	1.84
9	Isoquercitrin	C_21_H_20_O_12_	6.93	[M−H]^−^	463.0882	463.0871	−2.382	301	0.14
10	Syringaresinol glucoside	C_28_H_36_O_13_	7.78	[M−H]^−^	579.2083	579.2069	−2.419	417, 402, 181, 166	0.10

R_t_, retention time.

**Table 2 pharmaceuticals-17-01032-t002:** Effect of IP extract on prostatic weight and body weight.

Group	Treatment	Absolute Prostate Weight (g)	Body Weight (g)
NC	Corn oil/PBS	0.81 ± 0.13	436.67 ± 7.74
BPH	Testosterone/PBS	1.56 ± 0.04 ^##^	387.33 ± 26.21
FIN	Testosterone/Finasteride	1.07 ± 0.12 **	404.00 ± 60.67
IP	Testosterone/IP	1.35 ± 0.11 *	428.80 ± 24.02

Data are expressed as mean ± SD. ^##^ *p* < 0.01 compared with the NC group; * *p* < 0.05 and ** *p* < 0.01 compared with the BPH group. Body weight differences are not statistically significant between all groups.

## Data Availability

Data is contained within the article and Appendix A.

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
