# Peer review of "Ixeris polycephala Extract Alleviates Progression of Benign Prostatic Hyperplasia via Modification of Proliferation, Apoptosis, and Inflammation"

_pharmaceuticals, 2024, doi:10.3390/ph17081032_

Round 1

Reviewer 1 Report

Comments and Suggestions for Authors

The manuscript pharmaceuticals-3116436 reports the evaluation of ixeris polycephala in the treatment of benign prostatic hyperplasia. Based on my comments below I recommend the publication after minor revisions:

1) The legend of Table 1 provides the solvent used to extract the components of ixeris polycephala.

2) In the manuscript it is not clear the solvent used to extract the components of the plant. Perhaps they used polar solvent due to the types of compounds. Thus, explore in the manuscript if they can obtain other trend when they obtain extract by a non-polar solvent.

3) The authors identified by UPLC-MS/MS the main constituents of the extract. What is the ratio of the identified constituents? Please, provide it and explore if there is some role of the main component in the biological activity of the extract.

4) Please, provide the pharmacokinetic profile of the extract, i.e., area under the curve (AUC), maximum plasma concentration, time until Cmax is reached, volume of distribution, and half-life in the terminal phase. Correlate it with the obtained biological activity.

5) Please, provide the probability of survival (%) plot.

Reviewer 2 Report

Comments and Suggestions for Authors

- Review English

- The purpose of the study should be written clearly in the abstract

- What is the novelty of the study? I did not understand

- In lines 56-65. Explain the purpose and novelty of the study more comprehensively

- The resolution of the figures is low.

- I could not see the conclusion in the study. It should be added.

- The discussion is very weak, it should be expanded with current studies.

- Attention should be paid to abbreviations. A standard should be established.

Comments on the Quality of English Language

Good

Reviewer 3 Report

Comments and Suggestions for Authors

It is well known that herbs are used for a long time to treat human diseases. However, only a few have been in-depth investigated. In this manuscript, the therapeutic efficacy Ixeris polycephala extract against BPH was extensively studied. The findings in the experiment results were important, which disclose the potential application for treatment of BPH. In summary, this well-organized manuscript contained key findings was recommended to be published in this journal.

However, there are some issues in the Introduction and UPLC-MS/MS analysis to be addressed:

1. Different types of drugs for treating BPH were listed in the Introduction, however there were no specific representatives of these drugs. It is better to provide at least one example for each type of drugs, or references.

2. The references [9] and [10] reported the secondary metabolites of Ixeris sonchifolia, which was a different species from the title species Ixeris polycephala. Please revise them.

3. What did ‘Estimated’ represented in Table 1? Usually, the MS spectrum give the measured MS data, and calculated MS data was obtained from the molecular formula. And why there were no MS/MS fragments for compounds 4, 5 and 7?

4. As displayed in Figure 1, there were some main peaks were not identified, such as Rt ca. 12 min in the positive chromatograph. Please explain it.

Others:

1. Italic font for the species name ‘Ixeris polycephala’(P2L56), genus name ‘Ixeris’ (P2L56) and other term ‘Srd5a2’(P3L91).

2. The second half of the sentence ‘Testosterone is converted to DHT via 5α-reductase activity, and this enhanced level of DHT is a critical pathogenic mediator of BPH.’ was obscure to understand.

Comments on the Quality of English Language

Please check and revise the grammar and typo errors throughout the whole manuscript.
